# Analysis on Cascading Failures of Directed–Undirected Interdependent Networks with Different Coupling Patterns

**DOI:** 10.3390/e25030471

**Published:** 2023-03-08

**Authors:** Xiaojie Xu, Xiuwen Fu

**Affiliations:** Institute of Logistics Science and Engineering, Shanghai Maritime University, Shanghai 201306, China

**Keywords:** directed–undirected interdependent networks, cascading failures, coupling patterns, robustness, attack strategies

## Abstract

Most existing studies model interdependent networks as simple network systems consisting of two or more undirected subnets, and the interdependent edges between the networks are undirected. However, many real-world interdependent networks are coupled by a directed subnet and an undirected subnet, such as supply chain networks coupled with cyber networks, and cyber manufacturing networks coupled with service networks. Therefore, in this work, we focus on a ubiquitous type of interdependent network—the directed–undirected interdependent network—and research the cascading failures of directed–undirected interdependent networks with different coupling patterns. Owing to the diversity of coupling patterns to realistic interdependent network systems, we introduce two types of interdependent edges (i.e., directed-to-undirected and undirected-to-directed interdependent edges). On this basis, we generated different types of directed–undirected interdependent networks with varying coupling patterns (i.e., one-to-one, one-to-many, and many-to-one) and investigated the cascading failure robustness of these types of networks. Finally, we explored the cascading robustness of directed–undirected interdependent networks under two different attack strategies (single-node attack and multi-node attack). Through extensive experiments, we have obtained some meaningful findings: (1) the cascading robustness of directed–undirected interdependent networks is positively related to the overload tolerance coefficient and load exponential coefficient; (2) high-degree nodes and high-in-degree nodes should be protected to improve the cascading robustness of directed–undirected interdependent networks; (3) the cascading robustness of one-to-many interdependent networks can be improved by adding directed-to-undirected interdependent edges; and the cascading robustness of many-to-one interdependent networks can be improved by adding undirected-to-directed interdependent edges.

## 1. Introduction

Modern infrastructure systems are rarely isolated, but rather are highly interdependent [1,2,3,4]. Examples of such coupled critical infrastructures include power grids and water systems, information networks and power grids, seaports with information networks and airports, information networks and supply chains, etc. However, this interdependent pattern further exacerbates the risk of cascading failures in the system. Therefore, analyzing the impact of cascading failures on the network is essential for designing more robust interdependent network systems.

Until now, much research effort has been devoted to the analysis of cascading failures of interdependent systems [5,6,7,8,9,10,11,12,13,14]. Most of these studies modeled interdependent networks to be simple network systems consisting of two or more undirected networks, and the interdependent edges between the networks are undirected. However, many real-world interdependent networks are coupled by a directed subnet and an undirected subnet—e.g., supply chain networks coupled with cyber networks [15,16] and cyber manufacturing networks coupled with service networks [17,18]. Compared with the general interdependent network systems consisting of undirected networks, directed–undirected interdependent networks have significant differences in terms of the load redistribution and propagation process of cascading failure propagation processes. Unfortunately, there are relatively few investigators currently focused on this type of network. In addition, the current research mostly focuses on the cascading failure characteristics of the interdependent network under a single coupling pattern, and lacks the comparative analysis of the cascading failure characteristics of the interdependent network under different coupling patterns.

Therefore, this work focuses on the cascading failure characteristics of directed–undirected interdependent networks with different coupling patterns. First, we constructed a cascading failure model of directed–undirected interdependent networks. Then, we analyzed the impact of the coupling patterns on the cascading failure robustness of the interdependent networks. The main contributions of this work can be summarized as follows:

(1) We constructed an interdependent network model consisting of directed and undirected subnets with different coupling patterns. Considering the diversity of influence relationships between nodes at different levels in real-world interdependent network systems, this model uses two types of interdependent edges (i.e., directed edges pointing from the directed subnet to the undirected subnet and the directed edges pointing from the undirected subnet to the directed subnet).

(2) We investigated the cascading failure propagation process of the directed–undirected interdependent network with different coupling patterns (i.e., one-to-one coupling pattern, one-to-many coupling pattern, and many-to-one coupling pattern), which can provide support for understanding the impact of different coupling patterns on the cascading failure robustness of the network.

(3) We analyzed the robustness of interdependent networks under different attack strategies. We introduced two attack strategies (i.e., single-node attack and multi-node attack). The multi-node attack strategy can be further subdivided into six attack strategies (i.e., high-degree attack strategy, low-degree attack strategy, high-in-degree attack strategy, high-out-degree attack strategy, low-in-degree attack strategy, and low-out-degree attack strategy).

The rest of this paper is organized as follows. In Section 2, we review the related works. In Section 3, we develop a network model for directed–undirected interdependent networks with different coupling patterns. In Section 4, we introduce the cascading failure mechanism in detail. In Section 5, the simulation results are given. Finally, the paper is concluded.

## 2. Related Works

This section reviews the research progress on cascading failures of interdependent networks from two perspectives: subnet type and coupling pattern.

For the type of subnets, Buldyrev et al. [19] proposed a cascading failure model of power-communication interdependent network which was composed of undirected subnets and undirected interdependent edges. Yang et al. [20] constructed an electrical cyber-physical power network which is composed of two undirected subnets and analyzed the performance of interdependent network with two different coupling schemes. Mu et al. [21] proposed a realistic cascading model of the cyber-physical supply network to construct an information—physical-supply interdependent network, considering that overload of the cyber-power network triggers cascading failures and underload of the physical power supply network triggers chain failures. Wang et al. [22] proposed a high-level architecture (HLA)-based co-simulation method for critical infrastructure networks and established a case study of a dependent network consisting of two undirected subnets for two interdependent power and water systems. Tang et al. [23] analyzed the robustness of a one-to-one complex interdependent supply chain network consisting of a directed physical layer network and an undirected network layer network and developed a priority load redistribution strategy when modeling the network. Peng et al. [24] constructed a interdependent network model for industrial Internet of Things and analyzed the cascading failure dynamics of the intentional attack network. In this network model, the Industrial Internet of Things system is composed of IoT device networks and information networks. Parshani et al. [25] developed an interdependent network system composed of an interdependent worldwide seaport and airport networks and the worldwide airport network and proposed two quantities for measuring the level of inter-similarity between networks.

As for the coupling pattern, Parshani et al. [26] noted that reducing the number of interdependent edges could improve the robustness of the network to lead to a change from a first-order-percolation phase transition to a second-order-percolation transition. Zhong et al. [27] found that the repair effect will change as the proportion of dependent nodes increases. Wang et al. [28] proposed an assortative link, a disassortative link, and a random link by taking the characteristics of nodes on both sides of the interdependent edge into consideration and found that the assortative link was more robust. Shao et al. [29] pointed out that the coupling pattern of nodes in the actual interdependent networks was not one-to-one, but one-to-many or many-to-many. Then, they proposed an interdependent network model with multiple coupling patterns. To improve the robustness of the interdependent networks, Wang et al. [30] developed a coupling optimization strategy based on neighboring-node priority connection. In this strategy, the network degree distribution and the nodes which can avoid the propagation of failures are taken into account.

By observing Table 1, it is easy to find that most of the existing studies on real interdependent network systems focus on the simple network systems composed of two undirected subnets connected by undirected edges. In addition, the existing studies mostly focus on the cascading failure characteristics of the interdependent network under a single coupling pattern, and lack the comparative analysis of the cascading failure characteristics of the interdependent network under different coupling patterns.

## 3. Theoretical Model of the Directed-Undirected Interdependent Network

This section introduces the directed–undirected interdependent network model in detail. Table 2 shows the notation used in this model.

We consider an interdependent network model of two subnets as follows. The subnets of the directed–undirected interdependent network *G* are assumed to be a simple directed graph $G_{D}=\left\{V_{D},E_{D}\right\}$ and a simple undirected graph $G_{U}=\left\{V_{U},E_{U}\right\}$. In the directed graph $G_{D}$, $V_{D}=\left(v_{1}^{D},v_{2}^{D},\cdots v_{N_{D}}^{D}\right)$ and $E_{D}=\{\left(e_{i,j}^{D}|v_{i}^{D},v_{j}^{D}\in V_{D}\right\}$ denote the node set and directed edge set, respectively. In the undirected graph $G_{U}$, $V_{U}=\left\{\nu_{1}^{U},\nu_{2}^{U}\cdots ,\nu_{N_{U}}^{U}\right\}$ and $E_{U}=\left\{e_{i,j}^{U}|v_{i}^{U},v_{j}^{U}\in V_{U}\right\}$ denote the node set and undirected edge set, respectively. These two subnets, $G_{D}$ and $G_{U}$, are coupled via directed edges $E_{C}=\left\{e_{i,j}^{C}|v_{i}^{D}\in V_{D},v_{j}^{U}\in V_{U}\right\}$. The states of edges in $E_{D}$, $E_{U}$ and $E_{C}$ are defined by
(1)$$E_{i,j}^{D}=\left\{\begin{array}{cc} 0, & \mathrm{no}\;\mathrm{connection}\;\mathrm{from}\;\mathrm{node}\;v_{i}^{D}\;\mathrm{to}\;v_{j}^{U}\\ 1, & a\;\mathrm{directed}\;\mathrm{connection}\;\mathrm{pointing}\;\mathrm{from}\;\mathrm{node}\;v_{i}^{U}\;\mathrm{to}\;v_{j}^{D} \end{array}\right.,$$
(2)$$E_{i,j}^{U}=\left\{\begin{array}{cc} 0, & \mathrm{no}\;\mathrm{connection}\;\mathrm{from}\;\mathrm{node}\;v_{i}^{U}\;\mathrm{to}\;v_{j}^{U}\\ 1, & a\;\mathrm{undirected}\;\mathrm{connection}\;\mathrm{pointing}\;\mathrm{from}\;\mathrm{node}\;v_{i}^{U}\;\mathrm{to}\;v_{j}^{U} \end{array}\right.,$$
(3)$$E_{i,j}^{C}=\left\{\begin{array}{cc} 0, & \mathrm{no}\;\mathrm{connection}\;\mathrm{from}\;\mathrm{node}\;v_{i}^{D}\;\mathrm{to}\;v_{i}^{U}\\ 1, & a\;\mathrm{directed}\;\mathrm{connection}\;\mathrm{pointing}\;\mathrm{from}\;\mathrm{node}\;v_{i}^{D}\;\mathrm{to}\;v_{j}^{U}\\ -1, & a\;\mathrm{directed}\;\mathrm{connection}\;\mathrm{pointing}\;\mathrm{from}\;\mathrm{node}\;v_{j}^{U}\;\mathrm{to}\;v_{i}^{D} \end{array}\right..$$

In this network model, we consider the coupling patterns of subnets and the proportion of interdependent edges (completely interdependent and partially interdependent). We introduce the coupling patterns in detail. We consider three coupling patterns, i.e., one-to-one coupling, one-to-many coupling, and many-to-one coupling. In the one-to-one coupling pattern, a node in the undirected subnet $G_{U}$ can be connected to only one node in the directed subnet $G_{D}$ by an interdependent edge. The interdependent edge has two types, undirected-to-directed interdependent edges and directed-to-undirected interdependent edges, as shown in Figure 1. The undirected-to-directed interdependent edge is a directed edge from the undirected subnet $G_{U}$ to the directed subnet $G_{D}$, as shown in Figure 1a. The undirected-to-directed interdependent edge is a directed edge from the undirected subnet $G_{U}$ to the directed subnet $G_{D}$, as shown in Figure 1b. In the one-to-many coupling pattern, where “many” is set to *m*, a node in the directed subnet $G_{D}$ can be connected to m(m>1) nodes in the undirected subnet by interdependent edges. Figure 2 gives all possible combinations in the one-to-two coupling pattern. Figure 3 gives all possible combinations in the two-to-one coupling pattern.

To increase the flexibility of the network model, we define the numbers of nodes with interdependent edges in subnets $G_{D}$ and $G_{U}$ as $d_{D}$ and $d_{U}$, respectively. On this basis, we can easily get the proportions of nodes with interdependent edges $P_{D}$ and $P_{U}$ in subnets $G_{D}$ and $G_{U}$, respectively. In the one-to-one coupling pattern, $P_{D}$ = $P_{U}$. In the one-to-many coupling pattern where “many” is set to *m*, $P_{U}$ = $mP_{D}$. In the many-to-one coupling pattern where “many” is set to *m*, $P_{D}$ = $mP_{U}$. An example of the one-to-one interdependent network with $P_{D}$ = $P_{U}$ = 1 is shown in Figure 4.

Moreover, we assume that the average degree values of the two subnets are $\langle k_{D}\rangle$ and $\langle k_{U}\rangle$, respectively. We use $k_{i}^{D}=k_{i(in)}^{D}+k_{i(out)}^{D}$ to denote the degree of node, where $k_{i(in)}^{D}$ and $k_{i(out)}^{D}$ are the in-degree and out-degree of node $\nu_{i}^{D}$, respectively.

## 4. Cascading Failure Model of Directed–Undirected Interdependent Networks

### 4.1. Load and Capacity

In this section, we present a cascading model of directed–undirected interdependent networks, in which the network is limited by node capacity and the load is defined according to the degree of nodes.

In many practical directed network systems, the load of nodes is closely related to its out-degree. For example, in many service-oriented networks (e.g., supply chain networks), the out-degree of a node represents the number of service nodes that depend on the services it provides. Obviously, the higher the out-degree, the more services the node needs to provide, and the heavier its load. Therefore, in our model, the initial load of a node in the directed network is represented by its out-degree in an exponential way. We define the initial load of the node $v_{i}^{D}$ in the directed subnet $G_{D}$ as
(4)$$L_{i}^{D}\left(0\right)=k_{i(out)}^{D}{\left(0\right)}^{\beta },$$
where β>0 is the load-exponential coefficient to adjust the initial load of nodes. According to the “as much as required” principle [31], the nodes’ capacity is set to be positively related to their initial load:(5)$$C_{i}^{D}=\left(1+\sigma \right)L_{i}^{D}\left(0\right),$$
where σ≥0 is the overload-tolerance coefficient to indicate the capacity resources that the node has.

Similarly, the initial load of node $v_{i}^{U}$ in undirected subnet $G_{U}$ is defined as
(6)$$L_{i}^{U}\left(0\right)=k_{i}^{U}{\left(0\right)}^{\beta },$$
and the capacity of node $v_{i}^{U}$ is set to be linearly related to its initial load $L_{i}^{U}\left(0\right)$; that is,
(7)$$C_{i}^{U}=\left(1+\sigma \right)L_{i}^{U}\left(0\right).$$

### 4.2. Load-Redistribution Schemes

Existing studies generally assumed that the load on the failed node will be redistributed locally. Even redistribution schemes [32,33] and idle redistribution schemes [34,35] are the most widely used local load-redistribution schemes. Under these two schemes, the load initially on the failed node will be redistributed to its neighboring nodes. Since the idle capacity scheme can better help the network withstand the impact of the load redistribution process, we use this load-redistribution scheme in our model.

In directed subnet $G_{D}$, if node $v_{i}^{D}$ fails at time *t*, its out-degree neighboring node $v_{j}^{D}$ will receive extra load $\Delta L_{j}^{D}\left(t\right)$ from node $v_{i}^{D}$,:(8)$$\Delta L_{j}^{D}\left(t\right)=\frac{C_{i}^{D}-L_{i}^{D}\left(t\right)}{\sum_{j\in S_{i}^{D}\left(t\right)}\left[C_{j}^{D}-L_{j}^{D}\left(t\right)\right]}L_{i}^{D}\left(t\right),$$
where $S_{i}^{D}\left(t\right)$ is the set of the out-degree neighboring nodes of node $v_{i}^{D}$ at time *t*, and $C_{i}^{D}-L_{i}^{D}\left(t\right)$ is the spare capacity of node $v_{i}^{D}$ at time *t*. According to (8), under the idle redistribution scheme, nodes with more idle capacity will be assigned more load from the failed node.

To help understand the load-redistribution scheme, we present an example on a simplified network topology (shown in Figure 5). Assuming that node $v_{i}^{D}$ fails at time *t*, its load will transfer to its out-degree neighboring nodes according to the load-redistribution scheme. At time t+1, the loads of nodes $v_{b}^{D}$, $v_{c}^{D}$, and $v_{e}^{D}$ will update according to (9).
(9)$$\left\{\begin{array}{c} L_{b}^{D}\left(t+1\right)=L_{b}^{D}\left(t\right)+\Delta L_{b}^{D}\left(t\right)\\ L_{c}^{D}\left(t+1\right)=L_{c}^{D}\left(t\right)+\Delta L_{c}^{D}\left(t\right)\\ L_{e}^{D}\left(t+1\right)=L_{e}^{D}\left(t\right)+\Delta L_{e}^{D}\left(t\right) \end{array}\right..$$

If $L_{i}^{D}\left(t+1\right)>C_{i}^{D}$, and i∈{b,c,e}, node $v_{i}^{D}$ fails upon overload. Then, another round of node failures will be triggered. This cascading process will not stop until the loads of the remaining nodes are within their capacities.

In undirected subnet $G_{U}$, if node $v_{i}^{U}$ fails at time *t*, its neighboring node $v_{j}^{U}$ will receive extra load $\Delta L_{j}^{U}\left(t\right)$ from node $v_{i}^{U}$.
(10)$$\Delta L_{j}^{U}\left(t\right)=\frac{C_{i}^{U}-L_{i}^{U}\left(t\right)}{\sum_{j\in S_{i}^{U}\left(t\right)}\left[C_{j}^{U}-L_{j}^{U}\left(t\right)\right]}L_{i}^{U}\left(t\right),$$
where $S_{i}^{U}\left(t\right)$ is set of the neighboring nodes of node $\nu_{i}^{U}$ at time *t*, and $C_{i}^{U}-L_{i}^{U}\left(t\right)$ is the spare capacity of node $v_{i}^{U}$ at time *t*.

To illustrate the load redistribution process after a node failure in undirected subnet $G_{U}$ more clearly, we present an example on a simplified network topology (shown in Figure 6). Assuming that node $\nu_{i}^{U}$ fails at time *t*, the original load it takes will transfer to its neighboring nodes, $v_{b}^{U}$, $v_{c}^{U}$, $v_{d}^{U}$, and $v_{e}^{U}$, according to the load-redistribution scheme. At time t+1, the real time load of nodes $v_{b}^{U}$, $v_{c}^{U}$, $v_{d}^{U}$, and $v_{e}^{U}$ will update according to (11).
(11)$$\left\{\begin{array}{c} L_{b}^{U}\left(t+1\right)=L_{b}^{U}t+\Delta L_{b}^{U}\left(t\right)\\ L_{c}^{U}\left(t+1\right)=L_{c}^{U}t+\Delta L_{c}^{U}\left(t\right)\\ L_{d}^{U}\left(t+1\right)=L_{d}^{U}t+\Delta L_{d}^{U}\left(t\right)\\ L_{e}^{U}\left(t+1\right)=L_{e}^{U}t+\Delta L_{e}^{U}\left(t\right) \end{array}\right..$$

If $L_{i}^{U}\left(t+1\right)>C_{i}^{U}$, i∈{b,c,d,e}, node $v_{i}^{U}$ fails upon overload. Then, another round of node failures will be triggered, and this cascading process will not stop until the loads of the remaining nodes are within their capacities.

### 4.3. Cascading Mechanism

In the proposed cascading failure model, nodes will fall into failure due to three reasons: overload failure, isolation failure, and interdependency failure. The overload failure of a node refers to the failure caused by the load of node exceeding its capacity. The method for identifying isolation-failure node and interdependency-failure nodes will be described in detail below.

Unlike existing research models consisting of two undirected subnets, our model comprises a directed subnet and an undirected subnet. Therefore, in our model, the isolated failed nodes in the undirected subnet $G_{U}$ are the same as in existing studies; i.e., nodes that do not belong to the largest connected component. To illustrate the largest connected component more clearly, we present an example of a simplified network topology (shown in Figure 7). Assuming that node $v_{i}^{U}$ fails, the topology will be divided into two connected components. Connected component 1 consists of nodes $v_{a}^{U}$, $v_{b}^{U}$, $v_{c}^{U}$, and $v_{d}^{U}$; and connected component 2 consists of two nodes, $v_{e}^{U}$ and $v_{f}^{U}$. It is obvious that connected component 1 is the largest connected component. Therefore, the nodes in the connected component 2 fail due to isolation.

Similarly, for directed subnet $G_{D}$, if the node does not belong to the largest connected component, we consider this node to fall into isolation failure. It should be noted that there are two definitions of the largest connected component in the directed graph (i.e., largest weakly connected component and largest strongly connected component). Compared with the largest strongly connected component that requires two-way connectivity between two nodes, the largest weakly connected component is much looser and more suitable for observing the failure propagation process in directed networks. Therefore, the concept of the largest weakly connected component is commonly used to determine the size of the surviving nodes in directed networks [15,17]. In this work, we also use the largest weakly connected component to determine the connectivity of nodes in directed networks.

To illustrate the largest weakly connected component more clearly, we present an example of a simplified network topology (shown in Figure 8). Assuming that node $v_{i}^{D}$ fails, the topology will be divided into two weakly connected components. Any two of the four nodes $v_{a}^{D}$, $v_{b}^{D}$, $v_{c}^{D}$, and $v_{d}^{D}$ can have at least one directed path, so the four nodes $v_{a}^{D}$, $v_{b}^{D}$, $v_{c}^{D}$, and $v_{d}^{D}$ form weakly connected component 1. There is a directed path between nodes $v_{e}^{D}$ and $v_{f}^{D}$, which constitutes the weakly connected component 2. Apparently, weakly connected component 1 is the largest-weakly connected component. Hence, the nodes in weakly connected component 2 fail due to isolation.

In our model, the interdependent edges are directed. Hence, we define that the failure of the nodes at the start of the interdependent edges would lead to the failure of the nodes at the ends of the interdependent edges, whereas the failure of the nodes at the ends of the interdependent edges do not cause the failure of the nodes at the start of the interdependent edges. If node *i* is the end of the interdependent edges of *m* (m≥1) nodes, node *i* fails only when *m* nodes all fail; otherwise, node *i* works normally. If node *i* is the beginning of *m* (m≥1) interdependent edges, the failure or non-failure of *m* nodes will not stop node *i* from working properly. Figure 9 and Figure 10 each show an example of the interdependency failure, in a one-to-one and a one-to-two/two-to-one interdependent network, respectively.

### 4.4. Cascading Failure Mechanism of the Directed–Undirected Interdependent Network

Without loss of generality, in this part, we introduce the cascading failure mechanism by describing the cascading failure process triggered by removing a fraction of the nodes in the directed subnet $G_{D}$.

#### 4.4.1. Initial Stage of Cascading Failure Mechanism in the Directed–Undirected Interdependent Network

At the initial stage (t=1) of cascading failures, we perform the following steps.

Step 1: When a fraction *p* of nodes $V_{D}^{p}$ in directed subnet $G_{D}$ is removed, the load of failed nodes will propagate through connectivity link in the same subnet according to the responding load-redistribution scheme. Then, we check the load state of each node in the subnet, and put the nodes with loads exceeding their capacities into the overload failure set $V_{D}^{O}\left(1\right)$.

Step 2: Then, we check the connectivity state of the nodes in the directed subnet, $G_{D}$. If the nodes do not belong to the largest weakly connected component, we put these nodes into the isolation failure set $V_{D}^{I}\left(1\right)$.

Step 3: According to the interdependent edges of the failed node in $V_{D}^{F}\left(1\right)=V_{D}^{p}\cup V_{D}^{O}\left(1\right)\cup V_{D}^{I}\left(1\right)$, some nodes in undirected subnet $G_{U}$ fail due to loss of support nodes. We put these nodes into interdependency failure set $V_{U}^{E}\left(1\right)$.

Step 4: After that, we update the load redistribution of the undirected subnet $G_{U}$ and put the overload nodes into overload failure set $V_{U}^{O}\left(1\right)$. Then, we check the connectivity links and put the nodes that do not belong to the largest connected component into the isolation failure set, $V_{U}^{I}\left(1\right)$.

The framework of the initial stage of cascading failure mechanism in the directed–undirected interdependent network is shown in detail in Figure 11.

For the cascading failure propagation stage (t≥2), we perform the following steps.

Step 1: According to the interdependent edges of the failed nodes in $V_{U}^{F}\left(t-1\right)$, we find the node failures for interdependent edges in directed subnet $G_{D}$. Then, we put these failed nodes in the interdependency failure set, $V_{D}^{E}\left(t\right)$.

Step 2: We update the load redistribution of the directed subnet $G_{D}$ and put the overload nodes into overload failure set $V_{D}^{O}\left(t\right)$.

Step 3: Then, we check the connectivity links and put the nodes failure for isolation into the isolation failure set $V_{D}^{I}\left(t\right)$.

Step 4: We perform steps 1 to 3 on undirected subnet $G_{U}$ and get the interdependency failure set $V_{U}^{E}\left(t\right)$, overload failure set $V_{U}^{O}\left(t\right)$, and isolation failure set $V_{U}^{I}\left(t\right)$. After that, we can gain the latest failure node set $V_{U}^{F}\left(t\right)=V_{U}^{E}\left(t\right)\cup V_{U}^{O}\left(t\right)\cup V_{U}^{I}\left(t\right)$ in subnet $G_{U}$.

Step 5: We loop steps 1 to 4 until the failure node set $V^{F}\left(t\right)=V_{D}^{F}\left(t\right)\cup V_{U}^{F}\left(t\right)$ does not change.

The detailed framework of the cascading failure propagation stage in the directed–undirected interdependent network is shown in Figure 12.

#### 4.4.2. Example of Cascading Failure Process in a Directed–Undirected Interdependent Network

In this part, to help understand the proposed cascading failure mechanism in directed–undirected interdependent networks, we use a one-to-one interdependent network as an example for detailed explanation. As shown in Figure 13a, each subnet of the initial directed–undirected interdependent network has 10 nodes. Nodes in directed subnet $G_{D}$ are connected by directed connectivity links, and nodes in undirected subnet $G_{U}$ are connected by undirected connectivity links. As shown in Figure 13b, we first remove node $v_{6}^{D}$ of the directed subnet to trigger a cascading failure process. The original load of node $v_{6}^{D}$ is transferred to the out-degree neighboring node $v_{8}^{D}$, which causes node $v_{8}^{D}$ to fail due to overloading. Moreover, the connectivity links of the failed node are removed. As shown in Figure 13c, the failure of nodes $v_{6}^{D}$ and $v_{8}^{D}$ cause nodes $v_{4}^{D}$, $v_{5}^{D}$, $v_{9}^{D}$, and $v_{10}^{D}$ to fall into isolation failure. As shown in Figure 13d, the failure of node $v_{6}^{D}$ can affect the undirected subnet $G_{U}$ through an interdependent edge and cause the failure of node $v_{6}^{U}$. In addition, the load of node $v_{6}^{U}$ is transmitted to its neighboring nodes $v_{5}^{U}$ and $v_{9}^{U}$. We assume that node $v_{5}^{U}$ is not overloaded and that node $v_{9}^{U}$ failed via overload. Subsequently, the load of node $v_{9}^{U}$ propagates to the neighboring node $v_{10}^{U}$ and causes $v_{10}^{U}$ to fall into overload failure. As shown in Figure 13e, nodes $v_{7}^{U}$ and $v_{8}^{U}$ fail for do not belong to the largest connected component. By this step, the initial stage of the cascading failure process have been completed (Figure 13b–e). Starting from Figure 13f, the cascading failure process enters the next moment t=2. The failure of node $v_{7}^{U}$ affects the node $v_{7}^{D}$ through the interdependent edge and causes it to fail. Then, the load of node $v_{7}^{D}$ is transferred to node $v_{3}^{D}$ and causes the failure of node $v_{3}^{D}$. As can be seen in Figure 13g, node $v_{3}^{U}$ fails due to the failure of the support node $v_{3}^{D}$. Then, the load of node $v_{3}^{U}$ is transferred to its neighboring node $v_{4}^{U}$ and cause the failure of node $v_{4}^{U}$. However, the load redistribution of the load of node $v_{4}^{U}$ does not result in the failure of its neighboring node. As shown in Figure 13h, node $v_{5}^{U}$ fails to detach from the largest connected component. At this point, the cascading failure process will not continue, and the whole directed–undirected interdependent network has nodes $v_{1}^{D}$, $v_{2}^{D}$, $v_{1}^{U}$, and $v_{2}^{U}$ left. We can reasonably believe that the directed–undirected interdependent network has collapsed due to cascading failures.

#### 4.4.3. Time-Varying Interactive Equation of Cascading Failures

We construct time-varying interactive equations of cascading failures based on the cascading failure process described above. After the initial removal *P* fraction of the nodes, the remaining fraction of nodes in directed subnet $G_{D}$ is $\delta_{1}=1-p$. The load of the initial failed nodes will transfer to its out-degree neighboring nodes. Some nodes receiving an additional load may be overloaded. Therefore, the normal part of directed subnet $G_{D}$ is $O_{D}\left(\delta_{1}\right)$, where $O_{D}\left(\delta_{1}\right)$ is the fraction of the normal nodes after load redistribution in directed subnet $G_{D}$. After that, the overload failed nodes will disconnect some nodes from the largest weakly connected component. The normal fraction of directed subnet $G_{D}$ is $\delta_{1}^{{}^{\prime }}$ = $I_{D}\left(O_{D}\left(\delta_{1}\right)\right)$, where $I_{D}\left(O_{D}\left(\delta_{1}\right)\right)$ is the fraction of the normal nodes after chosing the nodes belong to the largest weakly connected component. As a fraction $q_{1}^{U}$ of nodes from undirected subnet $G_{U}$ depends on the failed nodes from directed subnet $G_{D}$, the fraction of the nodes that fail for interdependent edges is $q_{1}^{U}\left(1-\delta_{1}^{{}^{\prime }}\right)$. Accordingly, the normal nodes of undirected subnet $G_{U}$ are $\zeta_{1}=1-q_{1}^{U}\left(1-\delta_{1}^{{}^{\prime }}\right)$. The remaining fraction of nodes in undirected subnet $G_{U}$ is $\zeta_{1}^{{}^{\prime }}=I_{U}\left(O_{U}\left(\zeta_{1}\right)\right)$ after removing the overload-failed nodes and the nodes that disconnect to the largest connected component. As a fraction of $q_{2}^{D}$ nodes from directed subnet $G_{D}$ depends on the failed nodes from undirected subnet $G_{U}$, the fraction of the nodes that fail for interdependent edges is $q_{2}^{D}(N-\zeta_{1}^{{}^{\prime }})$. Accordingly, the normal nodes of directed subnet $G_{D}$ are $\delta_{2}=N-q_{2}^{D}(N-\zeta_{1}^{{}^{\prime }})$. After we remove the overload-failed nodes and the isolation-failed nodes, the normal fraction of nodes in directed subnet $G_{U}$ is $\delta_{2}^{{}^{\prime }}$ = $I_{D}\left(O_{D}\left(\delta_{2}\right)\right)$. It is easy to obtain the recursive relations of cascading failures according to the above description, and the time-varying iterative equations at each time can be summarized by
(12)$$\begin{array}{c} t=1:\left\{\begin{array}{c} G_{D}:\delta_{1}=1-p,\;\delta_{1}^{{}^{\prime }}=I_{D}\left(O_{D}\left(\delta_{1}\right)\right)\\ G_{U}:\zeta_{1}=1-q_{1}^{U}\left(1-\delta_{1}^{{}^{\prime }}\right),\;\zeta_{1}^{{}^{\prime }}=I_{U}\left(O_{U}\left(\zeta_{1}\right)\right) \end{array}\right.\\ t=2:\left\{\begin{array}{c} G_{D}:\delta_{2}=1-q_{2}^{D}(1-\zeta_{1}^{{}^{\prime }}),\;\delta_{2}^{{}^{\prime }}=I_{D}\left(O_{D}\left(\delta_{2}\right)\right)\\ G_{U}:\zeta_{2}=1-q_{2}^{U}\left(1-\delta_{2}^{{}^{\prime }}\right),\;\zeta_{2}^{{}^{\prime }}=I_{U}\left(O_{U}\left(\zeta_{2}\right)\right) \end{array}\right.\\ \vdots \\ t=n:\left\{\begin{array}{c} G_{D}:\delta_{n}=1-q_{n}^{D}(1-\zeta_{n-1}^{{}^{\prime }}),\;\delta_{n}^{{}^{\prime }}=I_{D}\left(O_{D}\left(\delta_{n}\right)\right)\\ G_{U}:\zeta_{n}=1-q_{n}^{U}\left(1-\delta_{n}^{{}^{\prime }}\right),\;\zeta_{n}^{{}^{\prime }}=I_{U}\left(O_{U}\left(\zeta_{n}\right)\right) \end{array}\right. \end{array},$$
where $\delta_{n}$ and $\zeta_{n}$ are the proportion of the normal nodes in the initial state of directed subnet $G_{D}$ and undirected subnet $G_{U}$ at iteration number *n*, respectively; $\delta_{n}^{{}^{\prime }}$ and $\zeta_{n}^{{}^{\prime }}$ are the proportion of the normal nodes after removing the overload failed nodes and the isolation failed nodes of directed subnet $G_{D}$ and undirected subnet $G_{U}$ at iteration number *n*, respectively. Equation (Equation 12) arrives at a steady state when n→∞, i.e., $\delta_{n}=\delta_{n}^{{}^{\prime }}=\delta_{n+1}=\delta_{n+1}^{{}^{\prime }}$ and $\zeta_{n}=\zeta_{n}^{{}^{\prime }}=\zeta_{n+1}=\zeta_{n+1}^{{}^{\prime }}$. At this moment, there are no more overload failed nodes isolation failed nodes and interdependency failed nodes in the interdependent network. Hence, we can obtain that
(13)$$n\to \infty :\left\{\begin{array}{c} G_{D}:\delta_{n}=1-q_{n}^{D}(1-\zeta_{n}^{{}^{\prime }})\\ G_{U}:\zeta_{n}=1-q_{n}^{U}\left(1-\delta_{n}^{{}^{\prime }}\right) \end{array}\right..$$
Finally, the normal nodes in directed subnet $G_{D}$ and undirected subnet $G_{U}$ are, respectively, $1-q_{n}^{D}(1-\zeta_{n}^{{}^{\prime }})$ and $1-q_{n}^{U}\left(1-\delta_{n}^{{}^{\prime }}\right)$.

## 5. Simulation Results

### 5.1. Simulation Setup

We used MATLAB to perform our simulation. MATLAB has the following advantages in simulating the cascading failures of directed–undirected interdependent networks: (1) through the built-in GUROBI optimizer, the time-consuming of simulation can be significantly reduced [36]; (2) the built-in complex network tool box can provide sufficient support for network modeling and statistics of experimental results. To capture the heterogeneity of many real-world networks, both subnets, $G_{D}$ and $G_{U}$, have scale-free characteristics. In the simulation, the directed subnet $G_{D}$ and the undirected subnet $G_{U}$ included 100 nodes each. The average degree of the directed subnet $G_{D}$ and the undirected subnet $G_{U}$ is $\langle k_{D}\rangle =\langle k_{U}\rangle =4$. In the proposed interdependent network model, there are two types of interdependent edges, undirected-to-directed interdependent edges and directed-to-undirected interdependent edges. In the experiments of the one-to-one interdependent network, the proportions of these two types of interdependent edges were consistent. The interdependent networks with one-to-many and many-to-one coupling patterns were generated by adding two types of interdependent edges to the one-to-one interdependent network. In the initial one-to-one interdependent network, the proportions of the nodes with interdependent edges in the directed subnet $G_{D}$ and undirected network $G_{U}$ were set to $P_{D}=P_{U}=0.2$.

### 5.2. Robustness Metrics

In the experiments, in order to fully evaluate the cascading robustness of the directed–undirected interdependent network, we introduced two attack strategies: a single-node attack strategy and a multi-node attack strategy. In the single-node attack strategy, we sequentially removed a node from the directed subnet or undirected subnet and observed the impact of cascading failures. We define the cascading robustness of the directed–undirected interdependent network under the single-node attack strategy as
(14)$$Q_{s}^{D}=\frac{\sum_{i\in V_{D}}\left(N_{i}^{D}+N_{i}^{U}\right)}{N_{D}\left(N_{D}+N_{U}\right)},$$
(15)$$Q_{s}^{D}=\frac{\sum_{i\in V_{U}}\left(N_{i}^{D}+N_{i}^{U}\right)}{N_{D}\left(N_{D}+N_{U}\right)},$$
where $Q_{s}^{D}$ is the cascading robustness of the directed–undirected interdependent network under single-node attack on the directed subnet; $N_{S_{i}}^{D}$ and $N_{S_{i}}^{U}$ represent the number of normal nodes in the directed subnet and undirected subnet after node $v_{i}^{D}$ is removed from the subnet; $V_{D}$ and $V_{U}$ are the sets of nodes in subnets $G_{D}$ and $G_{U}$. $Q_{s}^{D}$ is positively related to the cascading robustness of the directed–undirected interdependent network. $Q_{s}^{D}=1$ means that removing any node in the directed subnet will not trigger the cascading failures. In this case, we can consider that the entire interdependent network is immune to the single-node attacks on the directed subnet. In contrast, $Q_{s}^{D}=0$ means that removing any node in the directed subnet will paralyze the entire interdependent network due to cascading failures. Similarly, for $Q_{s}^{U}$, the above description also applies. In the single-node attacks, we do not need to consider the selection of attack objects, which allows us to focus on the impact of the network’s own attributes. Therefore, in the experiment, we mainly used the single-node attack strategy to evaluate the impacts of modeling parameters on the entire network system.

In the multi-node attack strategies, we remove a certain proportion *P* of nodes from one subnet. We used multiple commonly used attack strategies, i.e., high-degree attack (HDA), low-degree attack (LDA), high-in-degree attack (HiDA), high-out-degree attack (HoDA), low-in-degree attack (LiDA), and low-out-degree attack (LoDA). In the HDA strategy, high-degree nodes will be attacked first; in the LDA strategy, low-degree nodes will be attacked first; in the HiDA strategy, high-in-degree nodes will be attacked first; in the HoDA strategy, high-out-degree nodes will be attacked first; in the LiDA strategy, low-in-degree nodes will be attacked first; in the LoDA strategy, low-out-degree nodes will be attacked first. We define the cascading robustness of the directed–undirected interdependent networks under the multi-node attack strategy as
(16)$$Q_{m}^{D}=\frac{N_{D,D}^{P}+N_{U,D}^{P}}{N_{D}+N_{U}},$$
(17)$$Q_{m}^{U}=\frac{N_{D,U}^{P}+N_{U,U}^{P}}{N_{D}+N_{U}},$$
where $Q_{m}^{D}$ and $Q_{m}^{U}$, respectively, represent the cascading robustness levels of the directed–undirected interdependent network after multi-node attacks on the directed subnet and undirected subnets; $N_{U,D}^{P}$ represents the number of normal nodes in the undirected subnet (denoted by *U*) after removing proportion *P* of nodes in directed subnet (denoted by *D*). In the experiments, multi-node attack strategies were mainly used to analyze the main causes of network performance degradation during cascading failures.

### 5.3. Influences of Load-Redistribution Schemes

To evaluate the impacts of different load-redistribution schemes on the cascading robustness of directed–undirected interdependent networks, we compared the cascading robustness of directed–undirected interdependent networks under the idle redistribution scheme, the even redistribution scheme, and the degree redistribution scheme. The idle redistribution scheme was to distribute the load of the failed node to its neighboring nodes according to their spare capacities, the even redistribution scheme was to distribute the load of the failed node to its neighboring nodes equally, and the degree redistribution was to distribute the load of the failed node to its neighboring nodes according to their degree values.

Figure 14 shows the cascading robustness of directed–undirected interdependent networks with different load-redistribution schemes under the single-node attack. We can easily observe that the cascading robustness of the network under the idle redistribution strategy is significantly better than the other two redistribution strategies. This is due to the fact that under the idle redistribution strategy, nodes with more capacity can receive more load from the failed nodes, thereby minimizing the overload risk of nodes with insufficient capacities. In contrast, under the degree-redistribution strategy and the even-allocation strategy, nodes with insufficient capacity are not subject to addition protection, and may even receive more load from failed nodes, which will significantly increase the overload probability of these vulnerable nodes and lead to an increased risk of cascading failures in the network.

### 5.4. Influences of Modeling Parameters

In this section, we investigate the effects of network modeling parameters (i.e., the load-exponential coefficient β, the overload-tolerance coefficient σ, and the proportion of nodes with interdependent edges *P*) on the cascading robustness of directed–undirected interdependent networks. In studying the effects of β and σ in one-to-many and many-to-one networks, we used one-to-two and two-to-one interdependent networks as examples. As *P* is a concept that is specific to one-to-one interdependent networks. We only studied the effect of *P* on the cascading robustness in one-to-one interdependent networks.

As shown in Figure 15, as the overload-tolerance coefficient σ increases, the cascading robustness of directed–undirected interdependent networks can be greatly improved. This is because in our model, a higher σ means that nodes can have more capacity to cope with the load. In addition, we can also observe that in both single-node attack scenarios, an increase in the load-exponential coefficient β can significantly improve the cascading robustness of interdependent networks with different coupling patterns. Regarding the single-node attacks on the undirected subnet of the one-to-one interdependent network, in the cases of β=0.4 and σ=0.2, $Q_{s}^{U}=0.3$, which means that there is a high possibility of cascading failures. When σ stays the same and β rises to 2, the $Q_{s}^{U}$ is 0.7, which means that the risk and damage of cascading failures are greatly reduced. In addition, we can also observe that there is a threshold value $\beta^{*}$. When β is less than $\beta^{*}$, the increase in β is helpful to the cascading robustness of the interdependent network. When β is greater than or equal to $\beta^{*}$, the lifting effect of the parameter is saturated.

Figure 16 depicts the heatmaps of $Q_{s}^{D}$ and $Q_{s}^{U}$ in the parameter space [σ,β] under the single-node attack. We can easily observe a critical zone of [σ,β]. When σ and β fall into this zone, cascading failures will not occur. We call this zone the “safety zone”. By comparing the sizeS of the safety zone in two single-node attack scenarios, we can easily notice that the safety zone in undirected subnet is much larger than that in the directed subnet. This result shows that attacks on the directed subnet cause more damage to the entire interdependent network system than attacks on the undirected subnet.

According to Figure 17, the robustness of the directed–undirected interdependent network decreases as the proportion *P* of interdependent edges increases. It is easy to understand that an increase in the number of interdependent edges will further increase the coupling degree between the two subnets and thus increase the probability of a node falling into interdependency failure.

### 5.5. Influence of Coupling Patterns When Adding Different Interdependent Edges

We added specific types of interdependent edges to one-to-one interdependent networks to generate different types of one-to-many interdependent networks and many-to-one interdependent networks. On this basis, we analyzed the effect of different coupling patterns generated by adding different types of interdependent edges on the cascading robustness of the directed–undirected interdependent networks.

Figure 18 shows the effects of different coupling patterns on the cascading robustness of the interdependent network when increasing the directed-to-undirected interdependent edges. We can easily observe that in the one-to-many interdependent network, the cascading robustness decreases significantly as *m* increases, and in the many-to-one interdependent network, the cascading robustness can be significantly improved as *m* increases. This is due to the fact that in the one-to-many interdependent network, each node with interdependent edges in the directed subnet needs to establish connections with *m* nodes in the undirected subnet. In this case, the dependency of the undirected subnet on the directed subnet will gradually increase as the number of directed-to-undirected interdependent edges *m* increases, which makes the node failure in the directed subnet more easily propagated to the undirected subnet. In contrast, in the many-to-one interdependent network, each node with interdependent edges in the undirected subnet needs to establish connections with *m* nodes in the directed subnet. As the number of directed–undirected dependent edges *m* increases, the dependence of the undirected network on the directed subnet will gradually decrease, which makes it more difficult to propagate node failures of the undirected subnet to the directed subnet. At the same time, the conditions for node failure in the directed subnet will become more demanding.

Figure 19 shows the effects of different coupling patterns on the cascading robustness of the dependent network when increasing the undirected-to-directed dependent edges. Figure 18 shows the effects of different coupling patterns on the cascading robustness of the dependent network when increasing the undirected-to-directed dependent edges. It can be easily observed that the trend of cascading robustness with *m* for different coupling patterns is exactly the opposite of that of increasing the directed-to-undirected interdependent edges when increasing the undirected-to-directed interdependent edges. This is due to the fact that in the one-to-many interdependent network, as the number of undirected-to-directed interdependent edges *m* increases, the dependency of the directed subnet on the undirected subnet tends to decrease, which makes the propagation of node failures in the directed subnet to the undirected subnet become more difficult. In contrast, in many-to-one networks, the dependence of the directed subnet on the undirected subnet tends to increase as the undirected-to-directed interdependent edges *m* increase, which in turn leads to a significantly higher possibility of node failure propagation from the undirected subnet to the directed subnet.

### 5.6. Composition Analysis on Cascading Failures

To eliminate the effect of coupling patterns on the experimental results, we used one-to-one interdependent networks to explore the influences of six attack strategies on network robustness.

Figure 20 shows the cascading robustness when removing fraction *p* of nodes in the attacked subnet. The ratio *p* represents the proportion of nodes removed from the network in each multi-node attack according to the specific attack strategy. The higher the value of *p*, the greater the number of nodes removed from the network. We can easily find that cascading robustness gradually decreases as the attack ratio *p* increases. In addition, there are significant differences in the degrees of damage caused to the network by different attack strategies. In the case of attacks on the directed subnet, the most harmful to the network are the HiDA and HoDA strategies, followed by the LiDA and LoDA strategies. Among the attacks on the undirected subnet, the most damaging to the network is the HAD strategy, followed by the LDA strategy. Take the the multi-node attack scenario with undirected subnet as an example: Removing 20% of the nodes in the undirected subnet in the HDA strategy can crash the whole network. In contrast, even if 50% of the nodes in the undirected subnet are removed by the LDA strategy, more than 60% of the nodes in the entire network system can still work properly. This result tells us that to improve the cascading robustness of the directed–undirected interdependent network against malicious attacks, we should give priority to protecting the high-in-degree and high-out-degree nodes in the directed subnet and the high degree nodes in the undirected subnet.

Figure 21 depicts the composition of the failed nodes for the interdependent networks with different coupling patterns generated by adding directed-to-undirected interdependent edges. It can be easily observed that the highest proportion of failed nodes in the whole interdependent network system was overload-failure nodes, followed by isolation-failure nodes and interdependency-failure nodes. Therefore, we can improve the cascading robustness of the network by increasing σ and β to help the network nodes better withstand the influence of network-load redistribution. In addition, in the one-to-many interdependent network, the proportions of interdependency-failure nodes, overload-failure nodes, and isolation-failure nodes all increase as *m* increases; and in the many-to-one interdependent network, the proportions of interdependent failure nodes, overload-failure nodes, and isolation-failure nodes all decrease as *m* increases. The proportion of each type of faulty node varies with the same trend as *m*, due to the fact that the appearance of interdependency-failure nodes originates from the mutual propagation of node failures between subnets. Interdependency-failure nodes further affect the numbers of overload-failure nodes and isolation-failure nodes within the subnet due to load redistribution. Next, we explain the reason for the variation in the number of interdependency-failure nodes with *m* different coupling patterns by analyzing the coupling patterns that appear when adding directed-to-undirected interdependent edges.

In our model, the coupling patterns that occur when adding directed-to-undirected interdependent edges is shown in Figure 22. Figure 22a,b show the coupling patterns of one-to-many interdependent networks. In these coupling patterns, whether the directed or undirected subnet is under attack, the probability of a node in the other subnet failing due to interdependent edges increases with the increase in interdependent edges. For example, the failure of node $v_{1}^{D}$ in one-to-one coupling pattern causes node $v_{1}^{U}$ to fail, while the failure of node $v_{1}^{D}$ in one-to-five coupling pattern causes node $v_{1}^{U}$, $v_{2}^{U}$, $v_{3}^{U}$, $v_{4}^{U}$ and $v_{5}^{U}$ to fail. Figure 22c,d show the coupling patterns of many-to-one interdependent networks. In these coupling patterns, whether the directed or undirected subnet is under attack, the probability of a node in the other subnet failing due to interdependent edges decreases with the increase in interdependent edges. For example, in the one-to-one coupling mode, the failure of node $v_{1}^{D}$ causes $v_{1}^{U}$ to fail, whereas in the one-to-five coupling pattern, the failure of $v_{1}^{D}$, $v_{2}^{D}$, $v_{3}^{D}$, $v_{4}^{D}$ and $v_{5}^{D}$ leads to the failure of node $v_{1}^{U}$. Therefore, the results in Figure 22 can be easily understood.

Figure 23 depicts the composition of the failed nodes for the interdependent networks with different coupling patterns generated by adding undirected-to-directed interdependent edges. Just as increasing the directed-to-undirected dependence edges, the highest proportion of overload-failure nodes is followed by interdependency-failure nodes and isolation-failure nodes. Therefore, we can improve the cascading robustness of the network by increasing σ and β to help the network nodes better withstand the influence of network-load redistribution. In addition, in one-to-many and many-to-one interdependent networks, the trend of each type of failure node with *m* is opposite to the trend when adding directed-to-undirected interdependent edges. We will explain the above phenomenon by analyzing the coupling patterns that occur by increasing the undirected-to-directed interdependent edges.

Coupling patterns when adding undirected-to-directed interdependent edges are shown in Figure 24. Figure 24a,b are the coupling patterns of one-to-many interdependent networks. In these coupling patterns, whether the directed or undirected subnet is under attack, the probability of a node in the other subnet failing due to interdependent edges decreases as the interdependent edges increase in number. For example, the failure of node $v_{1}^{U}$ in one-to-one coupling pattern does not cause the node in $G_{D}$ to fail, whereas the failure of nodes $v_{2}^{U}$, $v_{3}^{U}$, $v_{4}^{U}$, and $v_{5}^{U}$ in a one-to-five coupling pattern causes node $v_{1}^{D}$ to fail. Figure 24c,d show the coupling patterns of many-to-one interdependent networks. In these coupling patterns, whether the directed or undirected subnet is under attack, the probability of a node in the other subnet failing due to interdependent edges increases as the interdependent edges increase in number. For example, in the one-to-one coupling mode, failure of $v_{1}^{D}$ leads to failure of $v_{1}^{U}$; and in the five-to-one coupling mode, failure of $v_{1}^{D}$ leads to failure of $v_{1}^{U}$, and further to failure of $v_{2}^{D}$, $v_{3}^{D}$, $v_{4}^{D}$ and $v_{5}^{D}$ in $G_{D}$. Therefore, the results in Figure 23 can be easily understood.

## 6. Conclusions

In this work, we developed a cascading failure model of directed–undirected interdependent networks with different coupling patterns to characterize its cascading failure characteristics. Through extensive experiments, we have identified the effects of key modeling parameters and coupling patterns on the cascading failure robustness of directed–undirected interdependent networks. Some lessons can be learned from the experimental results for building a robust directed–undirected interdependent network system:

(1) The cascading robustness of directed–undirected interdependent networks is positively related to the overload tolerance coefficient σ and load exponential coefficient β. In addition, there is a critical threshold $\beta^{*}$ for node load. When the load-exponential coefficient β is greater than or equal to $\beta^{*}$, the lifting effect of the parameter is saturated.

(2) The attack on directed subnets causes greater damage to the whole interdependent network system than that on undirected subnets. Therefore, the directed subnet should be protected with emphasis.

(3) High-degree nodes, high in-degree nodes and high-out-degree nodes should be protected to improve the cascading robustness of directed–undirected interdependent networks.

(4) The cascading robustness of one-to-many interdependent networks can be improved by adding directed-to-undirected interdependent edges; and the cascading robustness of the many-to-one interdependent networks can be improved by adding undirected-to-directed interdependent edges.

(5) The highest proportion of failed nodes in the whole interdependent network system is overload-failure nodes, followed by the isolation-failure nodes and interdependency-failure nodes. Therefore, we improve σ and β to help the network to withstand the influence of network-load redistribution.

Many studies have shown that the coupling preferences between interdependent networks can effectively improve the network cascading failure robustness by adjusting the coupling preferences [37,38]. In the next step of our work, we will design a cascading-robust topology evolutionary mechanism for directed–undirected interdependent networks based on coupling preferences.

## Figures and Tables

**Figure 1 entropy-25-00471-f001:**
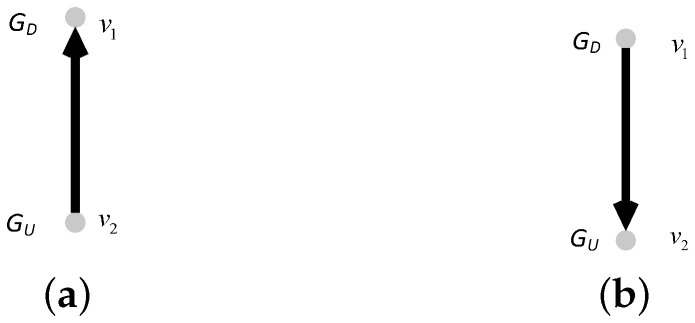
One-to-one coupling patterns between nodes in subnet $G_{D}$ and subnet $G_{U}$. (**a**) Undirected-to-directed interdependent edge. (**b**) Directed-to-undirected interdependent edge.

**Figure 2 entropy-25-00471-f002:**
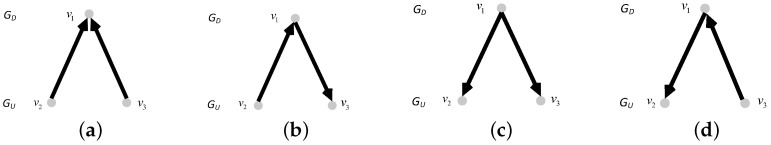
One-to-two coupling patterns between nodes in subnet $G_{D}$ and subnet $G_{U}$. (**a**) The one-to-two coupling pattern consisting of two undirected-to-directed interdependent edges. (**b**) The one-to-two coupling pattern consisting of an undirected-to-directed interdependent edge and a directed-to-undirected interdependent edge. (**c**) The one-to-two coupling pattern consisting of two directed-to-undirected interdependent edges. (**d**) The one-to-two coupling pattern consisting of a directed-to-undirected interdependent edge and an undirected-to-directed interdependent edges.

**Figure 3 entropy-25-00471-f003:**
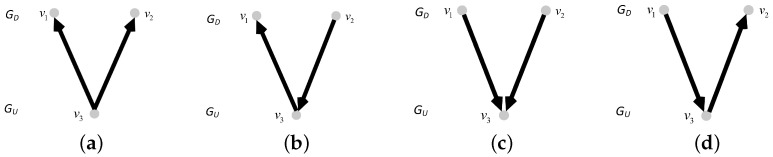
Two-to-one coupling patterns between nodes in subnet $G_{D}$ and subnet $G_{U}$. (**a**) The two-to-one coupling pattern consisting of two undirected-to-directed interdependent edges. (**b**) The two-to-one coupling pattern consisting of an undirected-to-directed interdependent edge and a directed-to-undirected interdependent edge. (**c**) The two-to-one coupling pattern consisting of two directed-to-undirected interdependent edges. (**d**) The two-to-one coupling pattern consisting of a directed-to-undirected interdependent edge and an undirected-to-directed interdependent edge.

**Figure 4 entropy-25-00471-f004:**
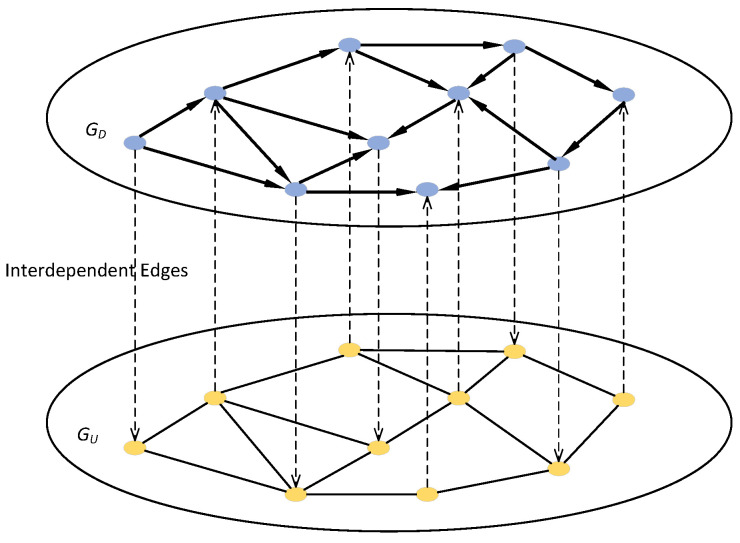
An example of a one-to-one interdependent network with $P_{D}$ = $P_{U}$ = 1.

**Figure 5 entropy-25-00471-f005:**
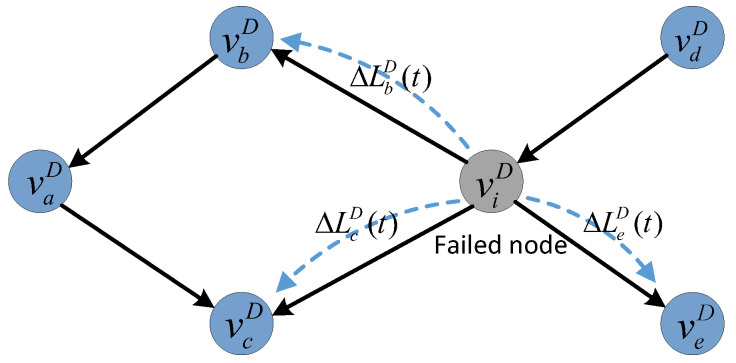
An example of load redistribution in the directed subnet $G_{D}$ after a node failure.

**Figure 6 entropy-25-00471-f006:**
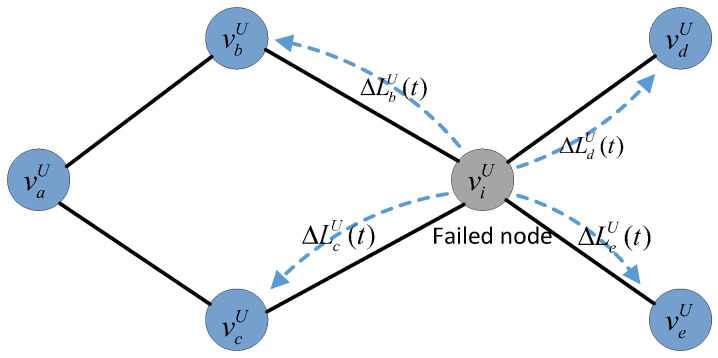
An example of load redistribution in the undirected subnet $G_{U}$ after a node failure.

**Figure 7 entropy-25-00471-f007:**
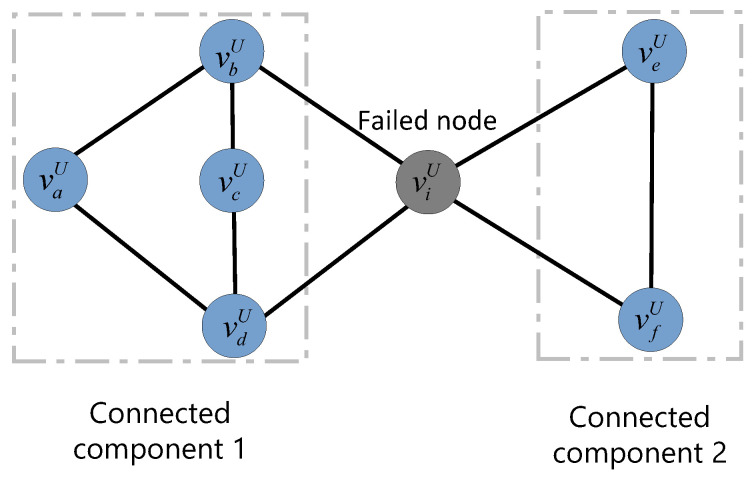
An example of the largest connected component in a undirected subnet after a node failure.

**Figure 8 entropy-25-00471-f008:**
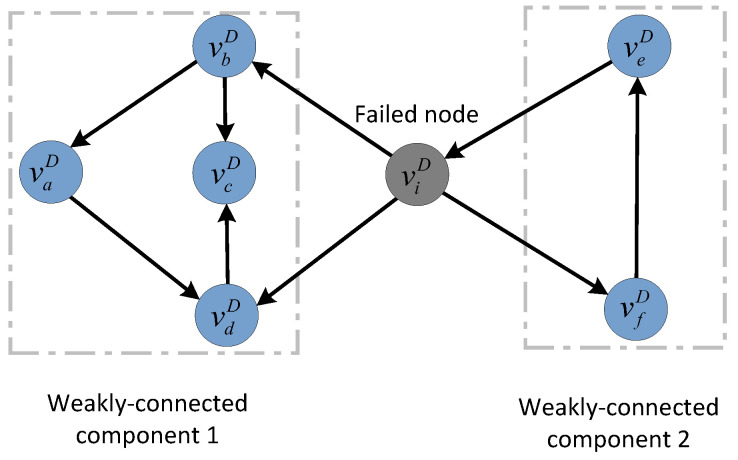
An example of the largest weakly connected component in a directed subnet after a node failure.

**Figure 9 entropy-25-00471-f009:**
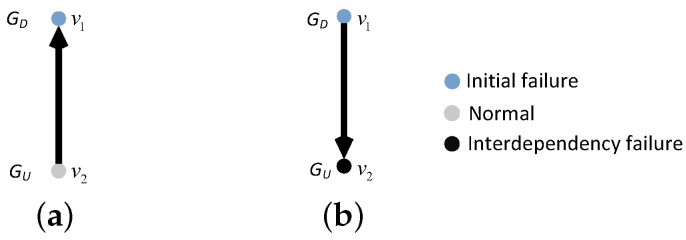
An example of interdependency failure in a one-to-one interdependent network. (**a**) Failure of node $v_{1}$ does not lead to failure of node $v_{2}$. (**b**)Failure of node $v_{1}$ leads to node $v_{2}$ interdependency failure.

**Figure 10 entropy-25-00471-f010:**
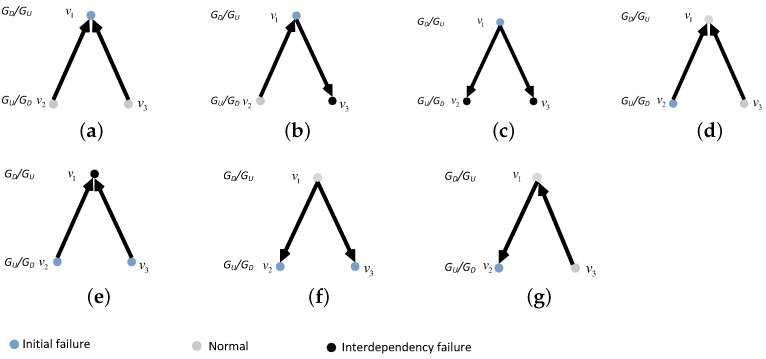
An example of interdependency failure in a one-to-two/two-to-one coupling interdependent network. (**a**) Failure of node $v_{1}$ does not lead to failure of node $v_{2}$ and $v_{3}$. (**b**) Failure of node $v_{1}$ lead to failure of node $v_{3}$. (**c**) Failure of node $v_{1}$ does lead to failure of node $v_{2}$ and $v_{3}$. (**d**) Failure of node $v_{2}$ does not lead to failure of node $v_{1}$. (**e**) Failure of node $v_{1}$ and $v_{2}$ lead to failure of node $v_{1}$. (**f**) Failure of node $v_{1}$ and $v_{2}$ does not lead to failure of node $v_{1}$. (**g**) Failure of node $v_{2}$ does not lead to failure of node $v_{1}$.

**Figure 11 entropy-25-00471-f011:**
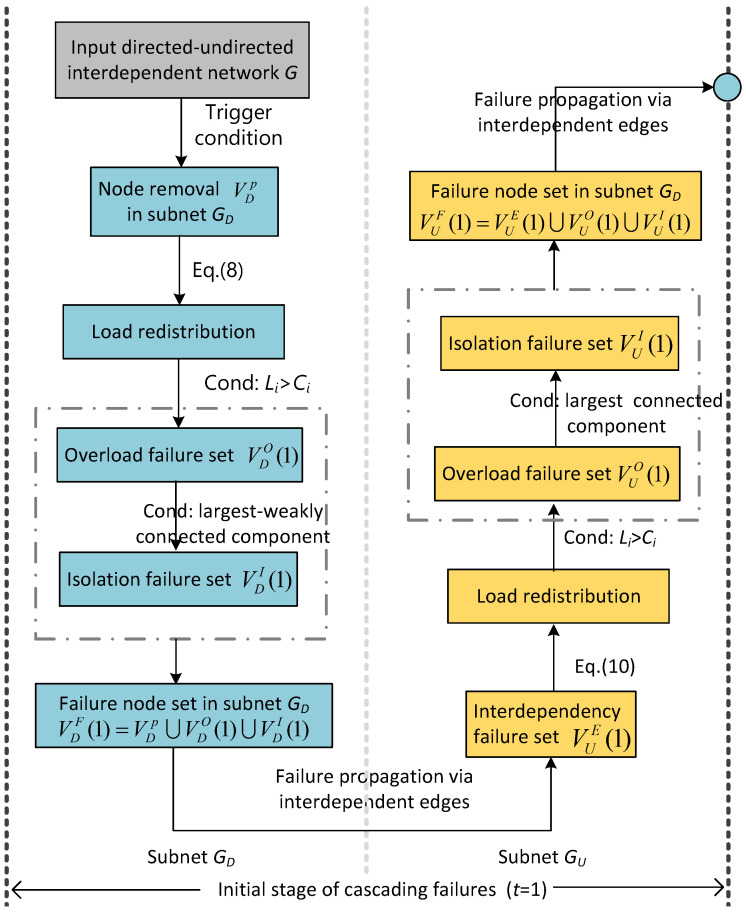
Initial stage of cascading failure mechanism in the directed–undirected interdependent network.

**Figure 12 entropy-25-00471-f012:**
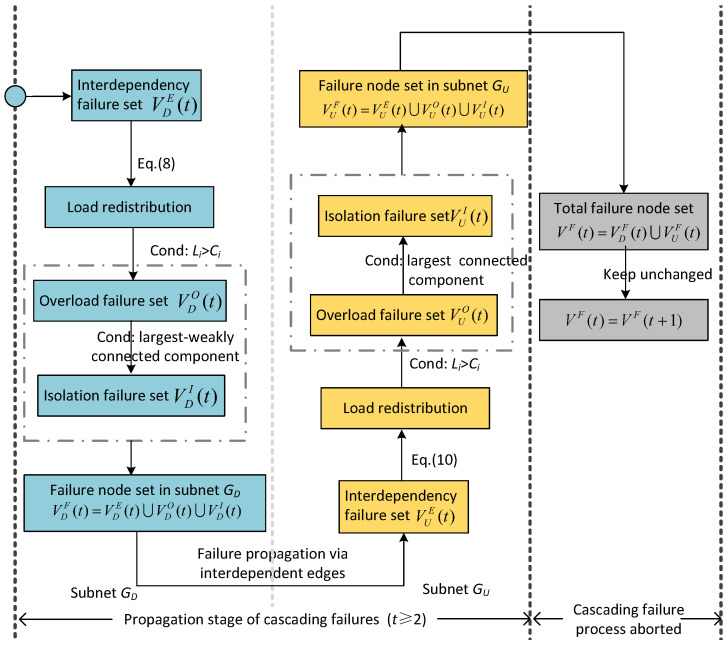
Propagation stage of cascading failure mechanism in the directed–undirected interdependent network.

**Figure 13 entropy-25-00471-f013:**
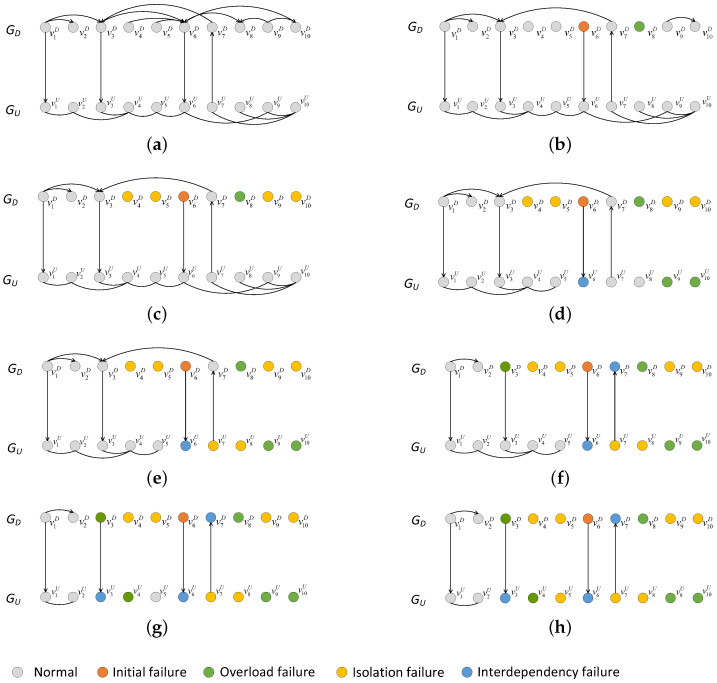
Cascading failure process in a one-to-one directed–undirected interdependent network. (**a**) Initial network. (**b**) Step 1. (**c**) Step 2. (**d**) Step 3. (**e**) Step 4. (**f**) Step 5. (**g**) Step 6. (**h**) Step 7.

**Figure 14 entropy-25-00471-f014:**
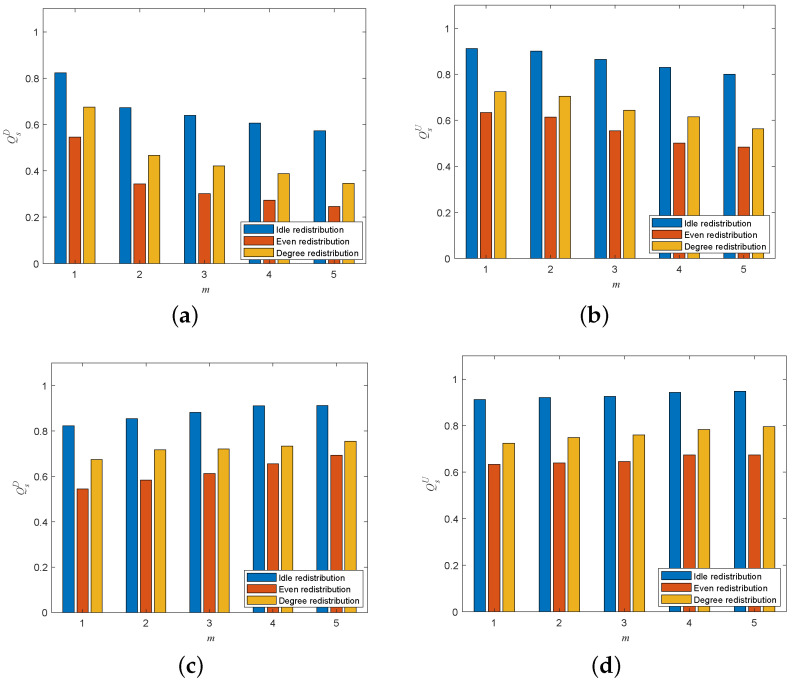
Cascading robustness of directed–undirected interdependent networks with different load-redistribution schemes under the single-node attack (σ=0.6, β=1). (**a**) Attack on the directed subnet of a one-to-many interdependent network. (**b**) Attack on an undirected subnet of a one-to-many interdependent network. (**c**) Attack on a directed subnet of a many-to-one interdependent network. (**d**) Attack on an undirected subnet of a many-to-one interdependent network.

**Figure 15 entropy-25-00471-f015:**
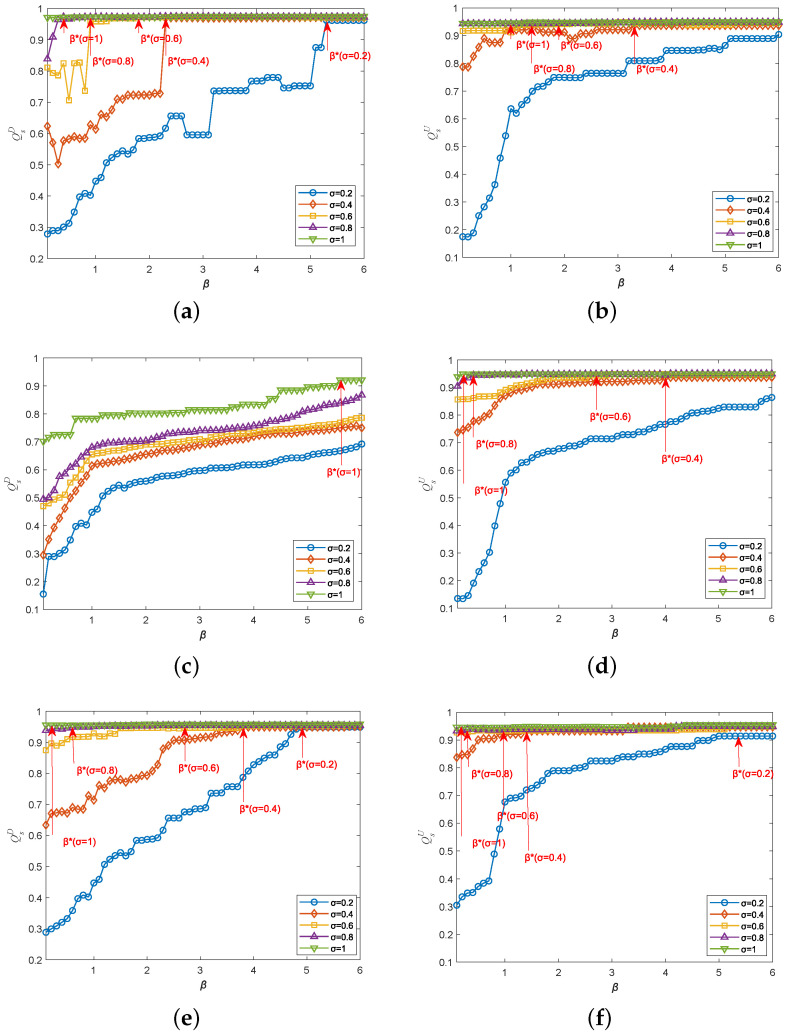
Cascading robustness while varying σ and β under the single-node attack. (**a**) Attack on the directed subnet of a one-to-one interdependent network. (**b**) Attack on the undirected subnet of a one-to-one interdependent network. (**c**) Attack on the directed subnet of a one-to-two interdependent network. (**d**) Attack on the undirected subnet of a one-to-two interdependent network. (**e**) Attack on the directed subnet of a two-to-one interdependent network. (**f**) Attack on the undirected subnet of a two-to-one interdependent network.

**Figure 16 entropy-25-00471-f016:**
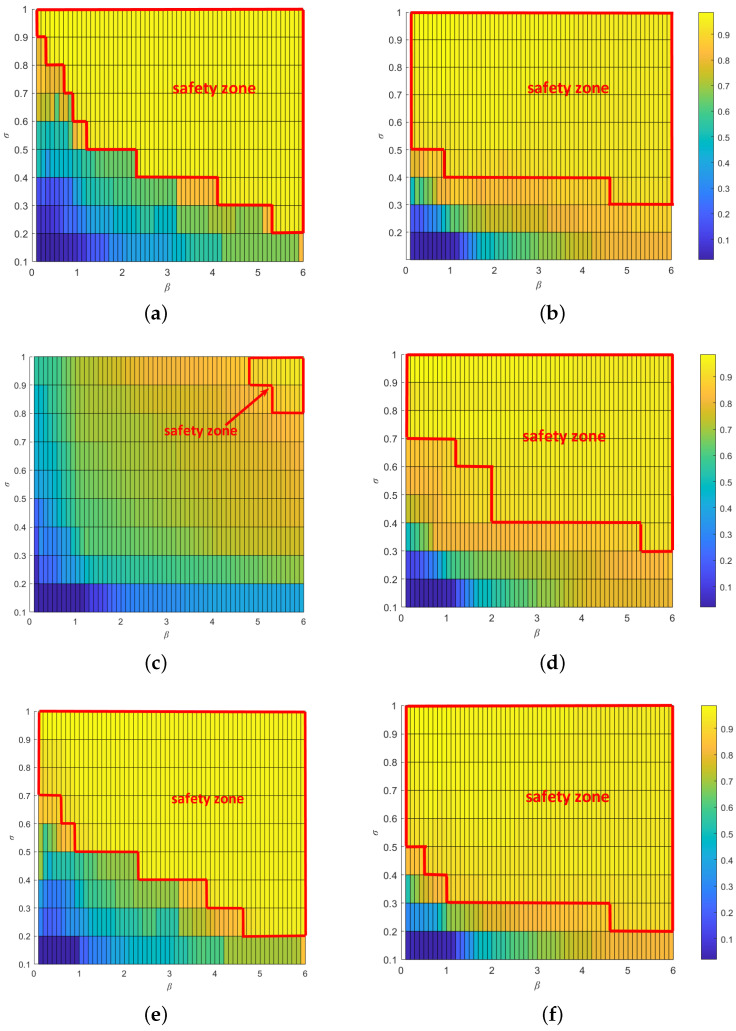
Heatmaps of $Q_{s}^{D}$ and $Q_{s}^{U}$ in the parameter space [σ,β] under the single-node attack. (**a**) Attack on the directed subnet of a one-to-one interdependent network. (**b**) Attack on the directed subnet of a one-to-one interdependent network. (**c**) Attack on the directed subnet of a one-to-two interdependent network. (**d**) Attack on the undirected subnet of a one-to-two interdependent network. (**e**) Attack on the directed subnet of a two-to-one interdependent network. (**f**) Attack on the undirected subnet of a two-to-one interdependent network.

**Figure 17 entropy-25-00471-f017:**
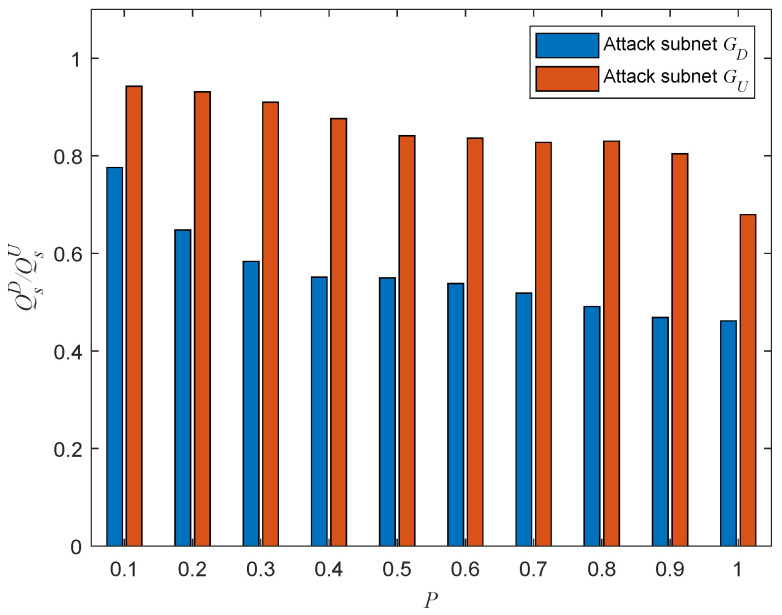
Network robustness with varying *P* under a single-node attack (σ=0.6, β=1).

**Figure 18 entropy-25-00471-f018:**
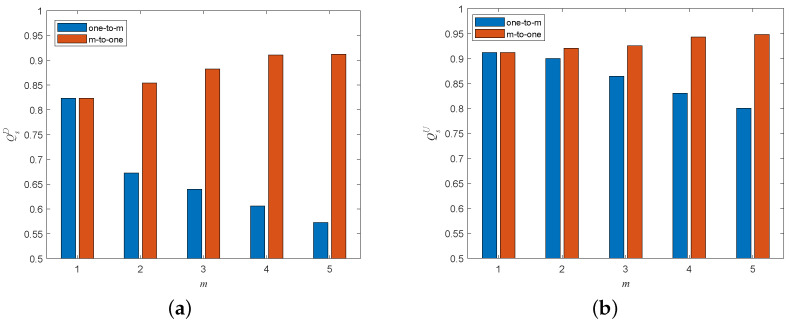
Network robustness with various coupling patterns when adding directed-to-undirected interdependent edges under a single-node attack (σ=0.6, β=1). (**a**) Attack on directed subnet. (**b**) Attack on undirected subnet.

**Figure 19 entropy-25-00471-f019:**
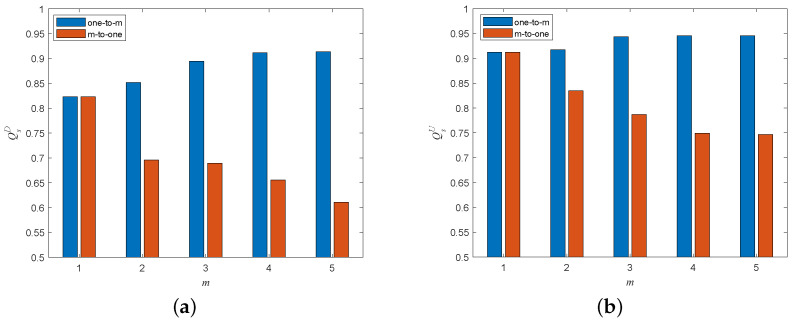
Network robustness with various coupling patterns when adding undirected-to-directed interdependent edges under the single-node attack (σ=0.6, β=1). (**a**) Attack on directed subnet. (**b**) Attack on undirected subnet.

**Figure 20 entropy-25-00471-f020:**
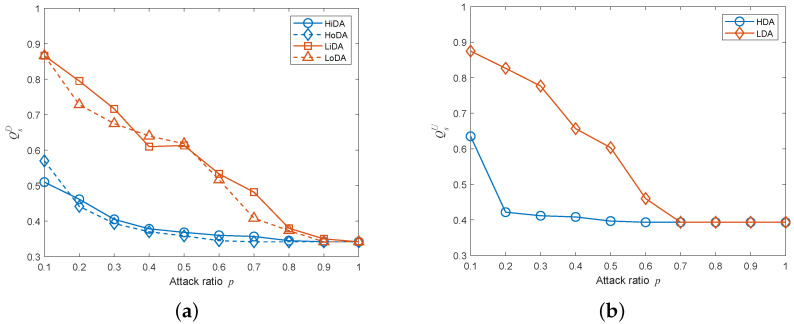
Network robustness under the different multi-node attack strategies (σ=0.6, β=1). (**a**) Attack on directed subnet. (**b**) Attack on undirected subnet.

**Figure 21 entropy-25-00471-f021:**
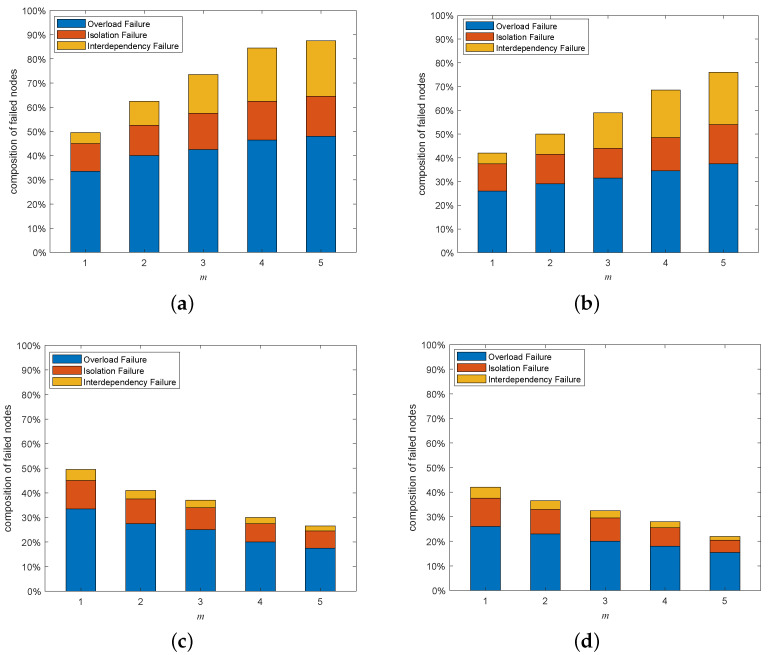
Composition of failed nodes when adding undirected-to-directed interdependent edges under a multi-node attack (σ=0.6, β=1, p=0.1). (**a**) HiDA on the directed subnet of a one-to-many interdependent network. (**b**) HDA on the undirected subnet of a one-to-many interdependent network. (**c**) HiDA on the directed subnet of many-to-one interdependent network. (**d**) HDA on the undirected subnet of many-to-one interdependent network.

**Figure 22 entropy-25-00471-f022:**
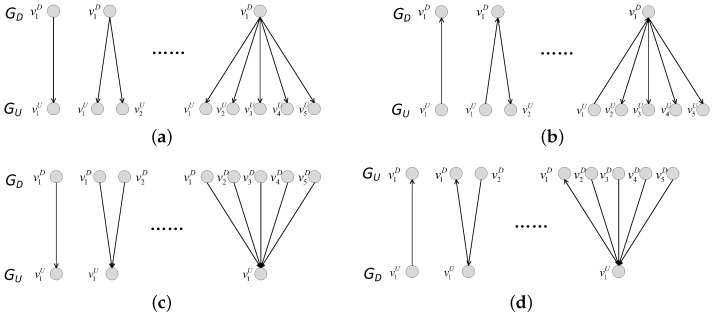
Coupling patterns when adding directed-to-undirected interdependent edges. (**a**) Coupling pattern 1 of one-to-many interdependent networks. (**b**) Coupling pattern 2 of one-to-many interdependent networks. (**c**) Coupling pattern 1 of many-to-one interdependent networks. (**d**) Coupling pattern 2 of many-to-one interdependent networks.

**Figure 23 entropy-25-00471-f023:**
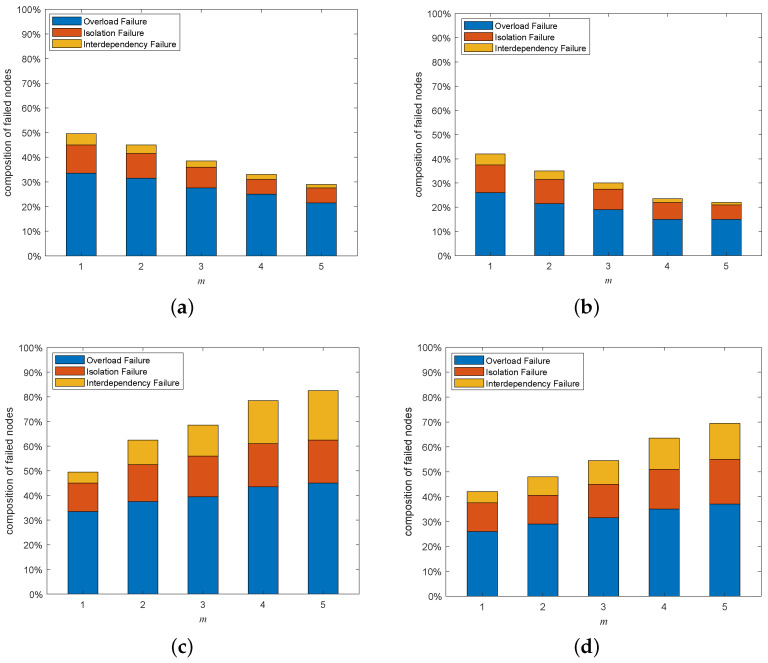
Composition of failed nodes when adding undirected-to-directed interdependent edges under the multi-node attack (σ=0.6, β=1, p=0.1). (**a**) HiDA on the directed subnet of a one-to-many interdependent network. (**b**) HDA on the undirected subnet of a one-to-many interdependent network. (**c**) HiDA on the directed subnet of a many-to-one interdependent network. (**d**) HDA on the undirected subnet of a many-to-one interdependent network.

**Figure 24 entropy-25-00471-f024:**
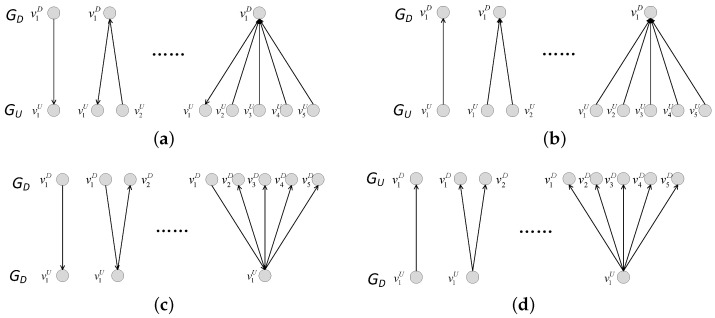
Coupling patterns when adding undirected-to-directed interdependent edges. (**a**) Coupling pattern 1 of one-to-many interdependent networks. (**b**) Coupling pattern 2 of one-to-many interdependent networks. (**c**) Coupling pattern 1 of many-to-one interdependent networks. (**d**) Coupling pattern 2 of many-to-one interdependent networks.

**Table 1 entropy-25-00471-t001:** Summary of existing interdependent network models.

Reference	Network Type	Coupling Pattern	Subnet Type	Interdependent Edge Type
[19]	Power-Information Network	one-to-one	Undirected-Undirected	Undirected
[20]	Power-Information Network	one-to-one	Undirected-Undirected	Undirected
[21]	Supply-Information Network	one-to-one	Undirected-Undirected	Undirected
[22]	Water-Power Network	one-to-one	Undirected-Undirected	Undirected
[23]	Supply-Information Network	one-to-one	Directed-Undirected	Undirected
[24]	IoT-Information Network	one-to-three	Undirected-Undirected	Undirected
[25]	Seaport-Airport- World-wide Airport Network	one-to-one	Undirected-Undirected-Undirected	Undirected
[26]	General Network	one-to-one	Undirected-Undirected	Undirected
[27]	General Network	one-to-one	Undirected-Undirected	Undirected
[28]	General Network	one-to-one	Undirected-Undirected	Undirected
[29]	General Network	one-to-one	Undirected-Undirected	Undirected
[30]	General Network	one-to-one	Undirected-Undirected	Undirected

**Table 2 entropy-25-00471-t002:** Notation in the directed–undirected interdependent network model.

Notation	Description
$G_{D}$	The directed subnet.
$G_{U}$	The undirected subnet.
$V_{D}$	The set of nodes in subnet $G_{D}$.
$V_{U}$	The set of nodes in subnet $G_{U}$.
$E_{D}$	The set of connectivity links in subnet $G_{D}$.
$E_{U}$	The set of connectivity links in subnet $G_{U}$.
$v_{i}^{D}$	A node i in directed subnet $G_{D}$.
$v_{i}^{U}$	A node i in directed subnet $G_{U}$.
$E_{C}$	Interdependent edge matrix between subnet.
$N_{D}$	Node number in subnet $G_{D}$.
$N_{U}$	Node number in subnet $G_{U}$.
$d_{D}$	Number of nodes with interdependent edges in subnet $G_{D}$.
$d_{U}$	Number of nodes with interdependent edges in subnet $G_{U}$.
$P_{D}$	Proportion of nodes with interdependent edges in subnet $G_{D}$.
$P_{U}$	Proportion of nodes with interdependent edges in subnet $G_{U}$.
*P*	Proportion of nodes with interdependent edges in both subnet of one-to-one interdependent networks.
$k_{i}^{D}$	Degree of node $v_{i}^{D}$.
$k_{i}^{U}$	Degree of node $v_{i}^{U}$.
$k_{i(in)}^{D}$	In-degree of node $v_{i}^{D}$.
$k_{i(out)}^{D}$	Out-degree of node $v_{i}^{D}$.
$\langle k_{D}\rangle$	Average degree of subnet $G_{D}$.
$\langle k_{U}\rangle$	Average degree of subnet $G_{U}$.
$S_{i}^{D}$	The set of out-degree nodes of node $v_{i}^{D}$.
$S_{i}^{U}$	The set of out-degree nodes of node $v_{i}^{U}$.
*p*	Proportion of nodes initially removed under multi-node attack.
$V_{D}^{p}$	The set of the initially removed nodes in subnet $G_{D}$.
$V_{D}^{O}\left(t\right)$	The set of overload failure nodes in subnet $G_{D}$ at time *t*.
$V_{D}^{I}\left(t\right)$	The set of isolation failure nodes in subnet $G_{D}$ at time *t*.
$V_{D}^{E}\left(t\right)$	The set of interdependency failure nodes in subnet $G_{D}$ at time *t*.
$V_{D}^{F}\left(t\right)$	The set of failure nodes in subnet $G_{D}$ at time *t*.
$V_{U}^{O}\left(t\right)$	The set of overload failure nodes in subnet $G_{U}$ at time *t*.
$V_{U}^{I}\left(t\right)$	The set of isolation failure nodes in subnet $G_{U}$ at time *t*.
$V_{U}^{E}\left(t\right)$	The set of interdependency failure nodes in subnet $G_{U}$ at time *t*.
$V_{U}^{F}\left(t\right)$	The set of failure nodes in subnet $G_{U}$ at time *t*.
$V^{F}\left(t\right)$	The set of failure nodes in directed–undirected interdependent network *G* at time *t*.

## Data Availability

Not applicable.

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
