# Peer review of "Analysis on Cascading Failures of Directed–Undirected Interdependent Networks with Different Coupling Patterns"

_entropy, 2023, doi:10.3390/e25030471_

Round 1

Reviewer 1 Report

This paper investigates the cascade failure dynamics of directed-undirected interdependent networks with different coupling modes and provides new insights into the behavior of these systems under stress.

The authors employ a combination of analytical methods and numerical simulations to model the behavior of directed-undirected interdependent networks with different coupling modes. The paper shows the effect of the load exponential coefficient and overload tolerance coefficient on the stability of the network, and improvement suggestions for optimizing the robustness of interdependent networks are given under the study of different attack strategies.

Maybe the following comments and suggestions can help to improve this paper for the readers.

1. This paper uses the idle redistribution scheme in the model. Please give a comparison under different Load redistribution schemes. Is there a different impact of different load redistribution schemes?

2. In Line 69 of the Introduction, the “low-degree attack strategy” is mentioned two times.

3. In part 4.3.3, please explain more clearly what is the symbol δ and δ′ exactly mean in your work, and also explain more clearly for the formulas 12 and 13.

4. Fig. 14 shows network robustness by varying σ and β under the single-node attack and one-to-one interdependent network, is it a specific type of interdependent network? Are there other limitations in this paper? Such as considering the Connection strategy.

5. In Conclusion 1, it is clearly stated that cascading robustness is related to the overload tolerance coefficient σ and the load exponential coefficient β. The conclusion is based on a one-to-one interdependent network. Is Conclusion 1 generally applicable under other coupling models? Interdependent networks with different topological coupling may have significant impacts on cascade failures.

6. Please give more descriptions of Figure 19. The meaning of the horizontal coordinate p should be presented.

Author Response

We have studied comments carefully and have made the corrections. Revised portions are marked in yellow in the paper. More details can be found in the attached response letter.

Reviewer 2 Report

1. Author need to mention their exact contribution.

2. Figure 1 need more detail explanation.

3. Algorithm need more detail explanation.

4. Experimental part need more clear provide

specific comments:

1. Why did author need to consider only two types direct and undirected subset edges ?

2.Author need to clearly discuss the cascade failure propagation process and their exact procedure .

3.why did author only consider coupling pattern one to one ?

4. Most of equation need to discuss in detail.

5.why did the node failure sure the weak directed subnet?

6. Author need to discuss algorithm and their step clearly .

7. Experimental part , author need to detail discuss each graph , experiments setting and parameters should be Clearly mention .

Author Response

(The authors gave the same response as above.)

Round 2

Reviewer 1 Report

In this paper, the authors study the cascade failure model of directed-undirected interdependent networks with different coupling modes, investigate the propagation process of a node failure, analyze the network robustness under different attack strategies, and propose improvement suggestions to optimize the robustness of interdependent networks.

After revising the comments submitted, the paper now has clearer data, more comprehensive figures as well as conclusions, and can be recommended for publication.